# MxB sensitivity of HIV-1 is determined by a highly variable and dynamic capsid surface

Richard J Miles[1], Claire Kerridge[1†], Laura Hilditch[1], Christopher Monit[1], David A Jacques[2], Greg J Towers[1]*

[1]Division of Infection and Immunity, University College London, London, United Kingdom; [2]EMBL Australia Node in Single Molecule Science and ARC Centre of Excellence in Advanced Molecular Imaging, School of Medical Sciences, University of New South Wales, Sydney, Australia

**Abstract** The type one interferon induced restriction factor Myxovirus resistance B (MxB) restricts HIV-1 nuclear entry evidenced by inhibition of 2-LTR but not linear forms of viral DNA. The HIV-1 capsid is the key determinant of MxB sensitivity and cofactor binding defective HIV-1 capsid mutants P90A (defective for cyclophilin A and Nup358 recruitment) and N74D (defective for CPSF6 recruitment) have reduced dependency on nuclear transport associated cofactors, altered integration targeting preferences and are not restricted by MxB expression. This has suggested that nuclear import mechanism may determine MxB sensitivity. Here we have use genetics to separate HIV-1 nuclear import cofactor dependence from MxB sensitivity. We provide evidence that MxB sensitivity depends on HIV-1 capsid conformation, rather than cofactor recruitment. We show that depleting CPSF6 to change nuclear import pathway does not impact MxB sensitivity, but mutants that recapitulate the effect of Cyclophilin A binding on capsid conformation and dynamics strongly impact MxB sensitivity. We demonstrate that HIV-1 primary isolates have different MxB sensitivities due to cytotoxic T lymphocyte (CTL) selected differences in Gag sequence but similar cofactor dependencies. Overall our work demonstrates a complex relationship between cyclophilin dependence and MxB sensitivity likely driven by CTL escape. We propose that cyclophilin binding provides conformational flexibility to HIV-1 capsid facilitating simultaneous evasion of capsid-targeting restriction factors including TRIM5 as well as MxB.

*For correspondence:
g.towers@ucl.ac.uk

Present address: †Department of Infectious Diseases, Kings College London, London, United Kingdom

Competing interests: The authors declare that no competing interests exist.

## Introduction

Mx proteins (MxA and MxB in humans [*Aebi et al., 1989*]) are interferon-induced, high molecular weight, GTPases with globular N-terminal domains forming an enzymatic active site and C-terminal leucine zipper domains connected by a hinge-like bundle-signalling element (BSE) (*Fribourgh et al., 2014*). MxA has well-characterised antiviral activity against influenza A virus (*Pavlovic et al., 1990*), *bunyaviruses* (*Kochs et al., 2002*) and *rhabdoviruses* (*Pavlovic et al., 1990*), whereas closely related MxB has been shown to inhibit HIV-1 infection (*Goujon et al., 2013*; *Kane et al., 2013*; *Liu et al., 2013*) and more recently hepatitis C virus (*Yi et al., 2019*) and *herpesviruses* (*Crameri et al., 2018*; *Schilling et al., 2018*). The antiviral mechanisms of Mx proteins is poorly understood. Cryo-electron microscopy of MxB has revealed higher order oligomers and large helical assembles (*Alvarez et al., 2017*). MxB dimerisation and oligomerisation appears to be important for anti-HIV activity, but larger helical assemblies are thought not to be required (*Alvarez et al., 2017*; *Buffone et al., 2015*; *Fricke et al., 2014*). Mx GTPase activity is highly conserved through evolution and is functional in human MxB (*King et al., 2004*), but surprisingly, is dispensable for MxB anti-HIV-1 activity (*Goujon et al., 2013*). MxA does not restrict HIV-1 but the transfer of the N-terminal nuclear

envelope targeting amino acids (1-26) from MxB to MxA generates a chimeric protein with anti-HIV-1 activity equivalent to the wild-type MxB protein (*Goujon et al., 2014*). This observation illustrates the dependence of MxB anti-HIV-1 activity on its amino terminal region and suggests that protein location at the nuclear membrane, conveyed by the MxB N terminus, is important for anti-viral activity. MxB is thought to suppress HIV-1 nuclear transport because it inhibits HIV-1 2-LTR circle formation but not viral DNA synthesis. Several studies have identified HIV-1 CA mutations that desensitise HIV-1 to MxB restriction (*Busnadiego et al., 2014*; *Goujon et al., 2013*; *Kane et al., 2013*; *Liu et al., 2013*; *Wei et al., 2016*), but biochemical CA-MxB binding studies have suggested that MxB escape mutations in the capsid do not prevent MxB interacting with CA in vitro (*Fribourgh et al., 2014*; *Fricke et al., 2014*; *Liu et al., 2015*; *Wei et al., 2016*). Recently, it has been suggested that MxB can interact with various components of the nuclear pore complex via its N-terminal domain (*Dicks et al., 2018*). Both Nup214 and TNPO1 were shown to be required for MxB nuclear envelope localisation, and for MxB restriction of HIV-1, through interaction with the triple arginine motif within the N-terminal domain of MxB. Together these observations suggest that MxB restricts HIV-1 nuclear entry through manipulation of the nuclear pore complex and/or transport machinery.

Here we sought to better understand the mechanism of MxB restriction and whether MxB sensitivity is dictated by the nuclear entry mechanism used by the virus. We find that the cellular cofactors used by HIV-1 to enter the nucleus, that is the nuclear entry mechanism, do not constitute a determinant of MxB sensitivity. We demonstrate conserved cofactor dependence for divergent wild type HIV-1 Gag proteins with different MxB sensitives and no effect of CPSF6 depletion on MxB sensitivity. We also show that the interaction of CypA with the classical reference strain R9 is essential for MxB sensitivity but that this interaction is not required for MxB sensitivity of a series of R9 mutants or divergent wild type HIV-1 Gag sequences. We propose that MxB sensitivity is determined by CA surface dynamics governed by CypA recruitment and CA sequence.

## Results

### MxB sensitivity is variable amongst primary viral isolates that are dependent on the same cofactors for nuclear import

Previous studies of MxB have suggested that MxB sensitivity is determined by CA sequence and influenced by changes in CA sequence driven by adaptive immune pressure, particularly from CTL escape (*Liu et al., 2015*; *Merindol et al., 2018*; *Wei et al., 2016*). The expression of MxB is tightly controlled by the type one interferon (IFN) - JAK-STAT axis (*Von Wussow et al., 1990*) and upon IFN stimulation, MxB expression is strongly induced (*Figure 1B*) and MxB proteins localize to diffuse cytoplasmic puncta and to a distinct ring in the region of the nuclear envelope, while the nucleus interior is largely devoid of MxB (*Figure 1A*). Since we found constitutive MxB expression to be toxic, and as we aimed to isolate the effect of MxB from that of other interferon-stimulated genes, we utilized a doxycycline (dox) inducible expression system to study the effects of MxB on HIV-1 infection. Importantly, induction of MxB expression in HEK293 cells with dox for 48 hr prior to infection, resulted in MxB levels similar to those seen after IFN-β induction (*Figure 1B*). Thus, in this system, MxB expression levels are similar to natural IFN-induced cellular levels. We then examined MxB sensitivity using VSV-G pseudotyped, HIV-1 virus like particles, bearing a GFP encoding genome (hereafter referred to as HIV-1 GFP). Our reference HIV-1 strain, here referred to as HIV-1 GFP R9 is made using the p8.91 packaging construct derived from the HIV-1 clone R9 (*Zufferey et al., 1997*). Importantly, HIV-1 R9 is identical to, or very closely related to, the viruses used in all other MxB studies.

As previously reported, MxB expression reduced the titre of HIV-1 GFP R9 by approximately 10-fold (*Figure 1C*; *Goujon et al., 2013*; *Kane et al., 2013*). Dox induced MxB antiviral activity against HIV-1 GFP R9 was consistent in HEK293, monocytic cell line THP-1 and T cell line A3.01 (*Figure 1C*). MxB expression in HEK293 cells did not effect HIV-1 GFP R9 reverse transcription (*Figure 1D*), but reduced 2-LTR circles and integrated proviral DNA (*Figure 1D*) as previously described (*Goujon et al., 2013*; *Kane et al., 2013*) consistent with suppression of viral nuclear entry. To probe MxB specificity we examined MxB sensitivity of a series of HIV-1 GFP constructs bearing *gag* genes from HIV-1(M) primary isolates originally selected to represent HIV-1(M) diversity (*Gao et al., 1998*;

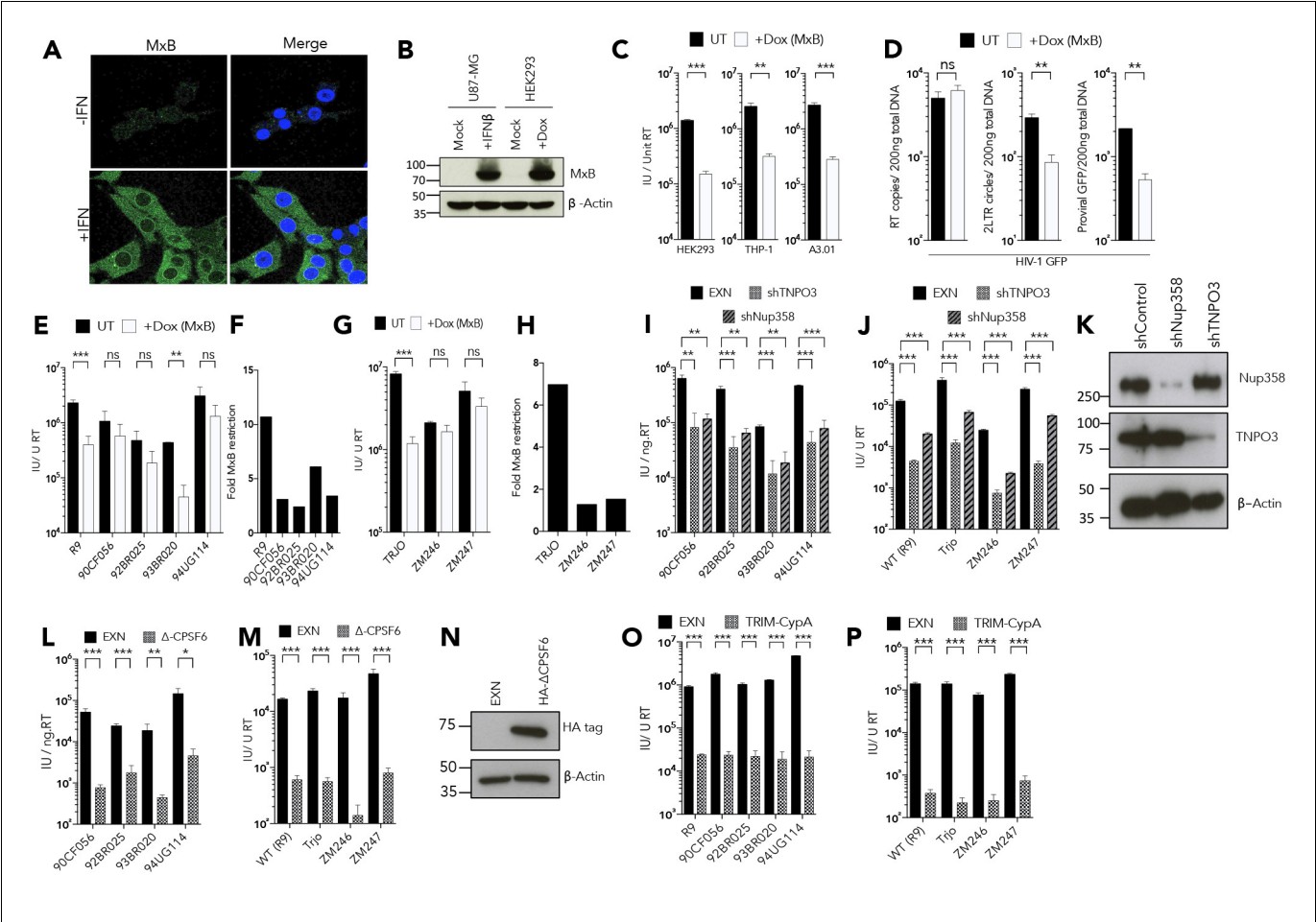

**Figure 1.** MxB restriction is independent of the viral nuclear import pathway. (a) Immunofluorescence staining MxB (green) in U87-MG cells 24 hr post-treatment with IFN (1000 U/ml). Counterstaining for DAPI (blue). (b) Immunoblot detecting MxB in U87-MG or HEK293 cell lysates after treatment with IFN or doxycycline as shown. Actin detected as a loading control. (c) HIV-1 GFP vector titre in HEK293-iMxB, THP-1-iMxB or A3.01-iMxB untreated (UT) or treated with Dox to induce MxB expression (+Dox (MxB)). (d) HIV-1 GFP RT products, 2LTR circles and proviral DNA of HIV-1 GFP vector in HEK293 untreated (UT) or treated with Dox to induce MxB expression (+Dox (MxB)). (e, g) Titre and (f, h) fold MxB restriction, of HIV-1 GFP vectors, containing gag-pro from. (e, f) founder isolates or. (g, h) other M-group clones, in HEK293 cells untreated (UT) or treated with Dox to induce MxB expression (+Dox (MxB)). Titre of HIV-1 GFP vectors, containing gag-pro sequences from the stated isolates, in. (i, j) HeLa cells stably expressing control, TNPO3 or Nup358 specific short hairpin RNA. (k) Matched representative immunoblot analysis from (i,j) detecting Nup358 or TNPO3. Actin detected as loading control. (l, m) HeLa cells stably expressing either control or deltaCPSF6. (n) Matched representative immunoblot analysis from (l,m) detecting HA epitope. Actin detected as loading control. (o, p) CRFK cells stably expressing either empty control vector or Trim-CypA. + / - SEM n = 3. (***) p<0.01 (**) p<0.05 (*) p<0.1 (ns) not significant p>0.1.

The online version of this article includes the following source data for figure 1:

**Source data 1.** Data presented in *Figure 4C*.

*Ikeda et al., 2004*; *Figure 1E*) or HIV-1(M) founder clones (*Salazar-Gonzalez et al., 2009*; *Figure 1G*). To our surprise we discovered quite different MxB sensitivities of between 10 fold and 2-fold inhibition, on MxB induction (*Figure 1F,H*). Given that only the nuclear periphery-localising long form of MxB restricts HIV-1 infection, and that MxB apparently restricts HIV-1 nuclear entry, we hypothesised that MxB sensitivity of the different Gag sequences might be influenced by their dependence on cofactors associated with nuclear entry. This notion is consistent with published demonstration that HIV-1 CA mutants, which have altered sensitivity to nuclear entry associated cofactor depletion, are insensitive to MxB restriction. For example, MxB-insensitive CA mutant N74D (*Goujon et al., 2013*; *Kane et al., 2013*) is insensitive to depletion of HIV-1 nuclear entry cofactors Nup358, TNPO3, CPSF6 and to some degree Nup153 (*Achuthan et al., 2018*; *Lee et al., 2010*; *Schaller et al., 2011*; *Sowd et al., 2016*). HIV-1 bearing CA P90A, which alters the otherwise

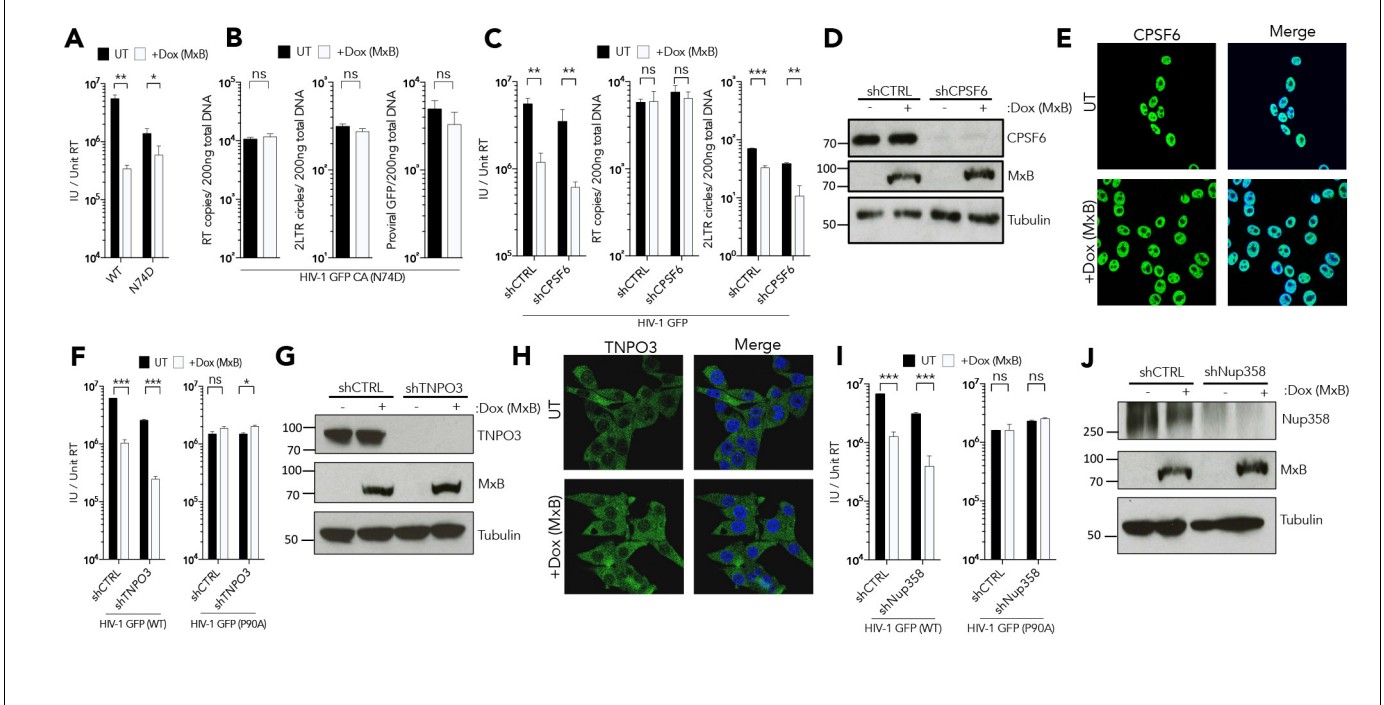

**Figure 2.** MxB restriction is independent of cofactor usage. (a) HIV-1 GFP WT or N74D vector titre on HEK293 cells untreated (UT) or treated with Dox to induce MxB expression (+Dox (MxB)). (a) HIV-1 GFP WT or N74D vector titre on HEK293 cells untreated (UT) or treated with Dox to induce MxB expression (+Dox (MxB)). (b) HIV-1 N74D RT products, 2LTR circles and proviral DNA in HEK293 untreated (UT) or treated with Dox to induce MxB expression (+Dox (MxB)). (c) HIV-1 GFP vector titre, RT products, 2LTR circles in HEK293 cells untreated (UT) or treated with Dox to induce MxB expression (+Dox (MxB)) and depleted of CPSF6 by shRNA transduction. (d) Matched representative immunoblot analysis from (c) detecting MxB or CPSF6. Tubulin detected as loading control. (e) Immunofluorescence of CPSF6 (yellow) in HEK293 cells untreated (UT) or treated with Dox to induce MxB expression (+Dox (MxB)). Counterstaining with DAPI (blue). (f) HIV-1 GFP WT (*left*) or P90A (*right*) vector titre on HEK293 cells untreated (UT) or treated with Dox to induce MxB expression (+Dox (MxB)) and depleted of TNPO3 by shRNA transduction. (g) Matched representative immunoblot analysis from (f) detecting MxB or TNPO3. Tubulin detected as loading control. (h) Immunofluorescence of TNPO3 (green) in HEK293 cells untreated (UT) or treated with Dox to induce MxB expression (+Dox (MxB)). Counterstaining with DAPI (blue). (i) HIV-1 GFP WT (*left*) or P90A (*right*) vector titre on HEK293 cells untreated (UT) or treated with Dox to induce MxB expression (+Dox (MxB)) and depleted of Nup358 by shRNA transduction. (j) Matched representative immunoblot analysis from (i) detecting MxB or Nup358. Tubulin detected as loading control. + / - SEM n = 3. (***) p<0.01 (**) p<0.05 (*) p<0.1 (ns) not significant p>0.1.

completely conserved CypA binding site and reduces CA affinity for CypA (*Yoo et al., 1997*) and Nup358 (*Schaller et al., 2011*), is completely insensitive to MxB restriction and also has reduced sensitivity to depletion of Nup358, TNPO3 and CPSF6. Both of these mutants also have retargeted proviral integration preferences consistent with altered nuclear entry mechanisms (*Achuthan et al., 2018*; *Lee et al., 2010*; *Schaller et al., 2011*; *Sowd et al., 2016*).

To probe the relationship between cofactor use and MxB sensitivity, we tested whether each of the divergent HIV-1 GFPs were sensitive to depletion of TNPO3 or Nup358. The karyopherin TNPO3 is a nuclear transport factor for CPSF6, and HIV-1 infectivity is reduced when TNPO3 is depleted in part due to the consequent relocalisation of CPSF6 to the cytoplasm (*Bejarano et al., 2019*; *De Iaco et al., 2013*; *Maertens et al., 2014*). Nup358 is a cytoplasmic component of the nuclear pore complex, thought to be required for HIV-1 to associate with the NPC via the Nup358 C-terminal Cyclophilin like domain (*Di Nunzio et al., 2012*; *Schaller et al., 2011*). However, viruses bearing both MxB sensitive, and insensitive, HIV-1(M) Gags had similar cofactor dependence, evidenced by similar sensitivities to stable depletions (*Figure 1K*) of TNPO3 or Nup358 in HeLa cells (*Figure 1I,J*). Depletion of HIV-1 cofactor CPSF6 does not typically cause a defect in infection in cell lines (*Lee et al., 2010*). We therefore tested the divergent Gag bearing HIV-1 GFP for sensitivity to restriction by a CPSF6 construct with a deleted C-terminal nuclear localisation signal (NLS) (*Rasaiyaah et al., 2013*). Again, HIV-1 GFP bearing each of the divergent Gags were sensitive to ΔCPSF6 expression (*Figure 1N*) consistent with CPSF6 use as a cofactor (*Figure 1L,M*). The CypA

binding site was conserved in all of the Gags tested and concordantly, each virus was sensitive to expression of a human TRIM5-CypA chimera (*Figure 1O and P*) suggesting all tested viruses are able to bind CypA. These observations demonstrated striking conservation of sensitivity to cofactor depletion and cofactor-derived restriction factors, and suggests that nuclear entry cofactor use does not dictate MxB sensitivity.

## Impact of CPSF6 and HIV-1 nuclear entry mechanism on MxB sensitivity

To further probe the mechanism of MxB restriction we returned to HIV-1 GFP R9 due to its particular sensitivity to MxB. As reported, HIV-1 GFP R9 bearing CA mutant N74D, which does not bind the cellular cofactor CPSF6, was insensitive to MxB (*Figure 2A,B*) and MxB induction did not strongly suppress infection (*Figure 2A*), viral DNA synthesis, or 2LTR circle formation (*Figure 2B*). To test whether HIV-1 CA N74D insensitivity is due to its different mechanism of nuclear entry, we measured the effect of CPSF6 depletion on MxB sensitivity of wild type HIV-1 GFP R9. This is informative because, like CA mutation N74D, CPSF6 depletion reduces HIV-1 R9 sensitivity to depletion of nuclear entry associated cofactors without impacting infectivity (*Lee et al., 2010*; *Ocwieja et al., 2011*; *Schaller et al., 2011*; *Sowd et al., 2016*). However, when we effectively depleted CPSF6 with shRNA (*Figure 2D*), HIV-1 remained sensitive to MxB (*Figure 2C*). In the absence of CPSF6, the MxB block to infection was accompanied by a reduction in 2LTR circles, as in wild type cells, suggesting MxB mediated suppression of nuclear entry was retained (*Figure 2C*). To our knowledge this is the first occasion in which the effects of CPSF6 depletion and CA N74D mutation on HIV-1 infection differ (*Lee et al., 2010*; *Ocwieja et al., 2011*; *Schaller et al., 2011*; *Sowd et al., 2016*). Importantly, CPSF6 was found in the nucleus by immunostaining 293 T cells and this was not altered by MxB expression suggesting that MxB does not inhibit HIV-1 by regulating CPSF6 traffic into the nucleus (*Figure 2E*). Together, these observations suggest that HIV-1 bearing CA N74D is insensitive to MxB as a direct result of the CA mutation and not due to loss of CPSF6 recruitment.

We also tested the effect of TNPO3 depletion on HIV-1 GFP R9 MxB sensitivity. Effective transient TNPO3 depletion (*Figure 2G*) reduced wild type HIV-1 GFP R9 infection in HEK293 cells but did not rescue infection from MxB (*Figure 2F*). TNPO3 depletion also did not affect the sensitivity of HIV-1 CA P90A to MxB, which was insensitive to TNPO3 depletion as described (*Schaller et al., 2011*; *Figure 2F*). MxB induction also did not relocalise TNPO3, suggesting that the MxB block to HIV-1 nuclear entry is direct and not due to MxB influencing TNPO3 nuclear transport. Since the longer, inhibitory form, of MxB is located around the nuclear rim, we tested whether Nup358 is required for MxB restriction. Transient Nup358 depletion did not significantly impact HIV-1 GFP infectivity in HEK293 cells (*Figure 2I*), despite efficient protein depletion (*Figure 2J*). Importantly, wild type HIV-1 GFP was fully sensitive to MxB after Nup358 depletion and HIV-1 P90A MxB insensitivity was also not affected (*Figure 2I*). Together, these observations suggest that MxB restriction of HIV-1 is independent of the cofactors used for nuclear entry, and therefore mutation of Gag appears to be impacting MxB sensitivity in a more direct way.

## Impact of cyclophilin A on MxB sensitivity

We next sought to understand how cyclophilin A impacts HIV-1 sensitivity to MxB. HIV-1 bearing CA mutant P90A, which alters the otherwise completely conserved CypA binding site and causes reduced interaction with CypA (*Yoo et al., 1997*) and Nup358 (*Schaller et al., 2011*), is completely insensitive to MxB restriction. But HIV-1 P90A also has reduced sensitivity to depletion of Nup358, TNPO3 and CPSF6 and has retargeted proviral integration preferences, which again suggests a connection between nuclear transport and MxB sensitivity (*Achuthan et al., 2018*; *Lee et al., 2010*; *Schaller et al., 2011*; *Sowd et al., 2016*).

We verified that HIV-1 CA P90A is insensitive to MxB demonstrating no reduction in HIV-1 P90A infectivity (*Figure 3A*), 2LTR or proviral DNA levels (*Figure 3B*) after induction of MxB expression. Furthermore, we confirmed the observation (*Liu et al., 2013*) that CypA depletion, here using shRNA (*Figure 3D*), fully rescues wild type HIV-1 GFP from MxB restriction, leading to a recovery of 2LTR circles and infectivity (*Figure 3C*). Addition of the CypA inhibitor, cyclosporine A (CsA), at the time of infection, also rescued wild type HIV-1 infectivity from MxB restriction (*Figure 3E*), consistent with CypA recruitment by CA being necessary for MxB sensitivity. We also found that HIV-1 infectivity can be almost completely rescued from MxB restriction, by the addition of CsA, up to 8 hr post

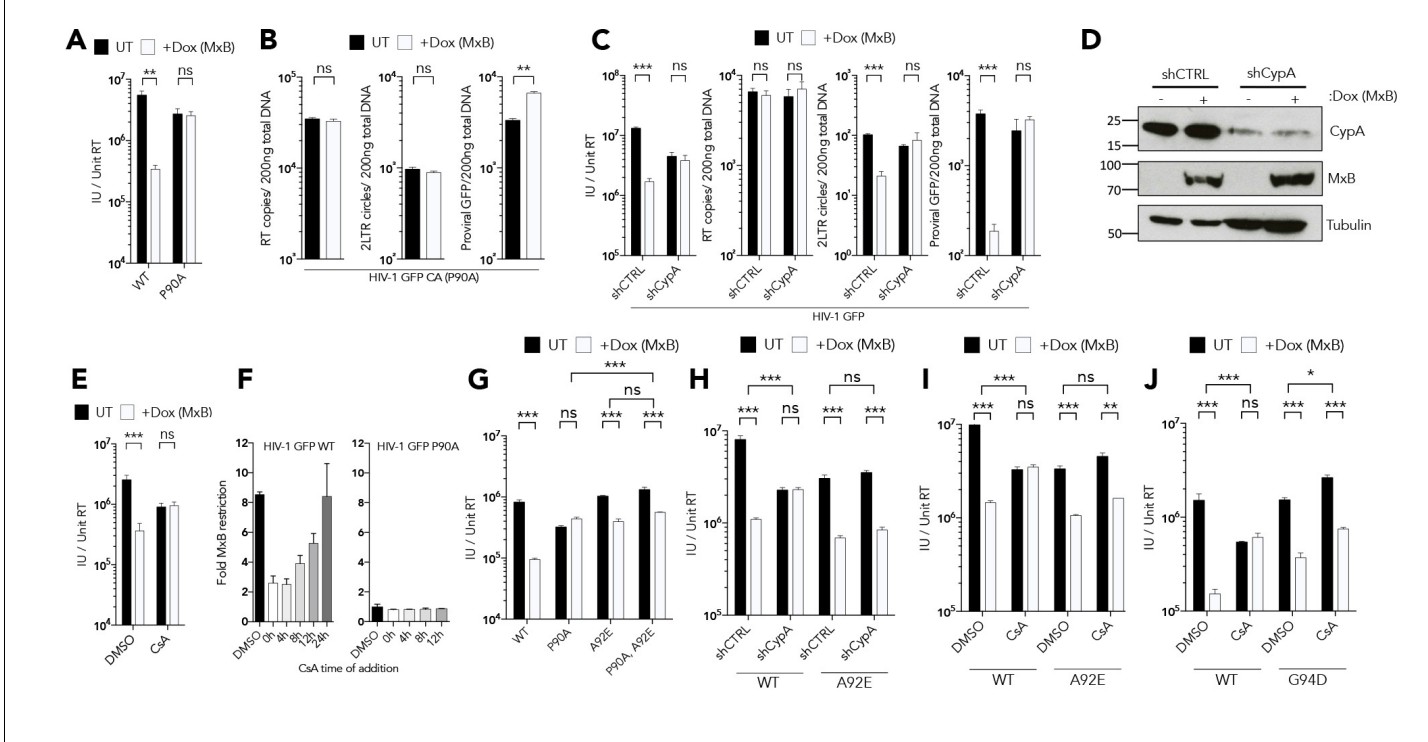

**Figure 3.** CypA is required for MxB restriction. (**a**) HIV-1 GFP WT or P90A vector titre in HEK293 untreated (UT) or treated with Dox to induce MxB expression (+Dox (MxB)). (**b**) HIV-1 GFP P90A RT products, 2LTR circles and proviral DNA in HEK293 untreated (UT) or treated with Dox to induce MxB expression (+Dox (MxB)). (**c**) HIV-1 GFP vector titre, RT products, 2LTR circles in HEK293 cells untreated (UT) or treated with Dox to induce MxB expression (+Dox (MxB)) and depleted of CypA by shRNA transduction. (**d**) Matched representative immunoblot analysis from (**c**) detecting MxB or CypA. Tubulin detected as loading control. (**e**) HIV-1 GFP vector titre on HEK293 untreated (UT) or treated with Dox to induce MxB expression (+Dox (MxB)) after treatment with CsA at the point of infection. (**f**) HIV-1 GFP WT (*left*) or P90A (*right*) vector titre on HEK293 cells untreated (UT) or treated with Dox to induce MxB expression (+Dox (MxB)) after CsA was added at the stated times after infection. Titre of HIV-1 GFP vector containing the stated CA mutants in. (**g**) HEK293 cells untreated (UT) or treated with Dox to induce MxB expression (+Dox (MxB)),. (**h**) HEK293 cells untreated (UT) or treated with Dox to induce MxB expression (+Dox (MxB)) and depleted of CypA by shRNA transduction or. (**i, j**) HEK293 cells untreated (UT) or treated with Dox to induce MxB expression (+Dox (MxB)) after treatment with CsA at the point of infection. + / - SEM n = 3. (***) p<0.01 (**) p<0.05 (*) p<0.1 (ns) not significant p>0.1.

infection, with the efficacy of CsA diminishing 24 hr post infection (*Figure 3F*). This shows that inhibition by MxB is reversible and suggests that the interaction between HIV-1 and MxB is not destructive, at least not immediately (*Figure 3F*). In a control experiment, HIV-1 CA P90A was treated with CsA during infection of MxB expressing cells and we observed no rescue to infectivity, consistent with P90A being resistant to MxB due to failure to recruit CypA (*Figure 3F*). Together, these data are consistent with CypA either acting as a cofactor for MxB interaction with the HIV-1 capsid or MxB targeting a form or conformation of Gag that is CypA dependent.

To distinguish between these two possibilities, we combined mutation A92E with P90A on HIV-1 CA and examined MxB sensitivity. The A92E mutation was identified by selection of HIV-1 replicating mutants during CypA inhibition with CsA (*Aberham et al., 1996*). This mutation appears to recapitulate the effect of CypA binding by reducing the dynamics of the CypA binding loop on the surface of the CA (*Lu et al., 2015*) leading to core stabilisation (*Li et al., 2000*). Thus, HIV-1 bearing CA A92E infects HEK293 cells efficiently, in the absence of CypA recruitment, enabling us to test whether CypA recruitment, or the conformational effect of CypA binding, is required for MxB sensitivity. In fact, adding A92E to HIV-1 CA P90A rescued sensitivity to MxB. Thus, MxB induction with dox restricted HIV-1 CA P90A A92E but not HIV-1 bearing CA P90A alone (*Figure 3G*). Importantly, the A92E CA mutation also rescued MxB sensitivity in the absence of CypA protein, after shRNA-mediated CypA depletion, and after CypA inhibition with CsA (*Figure 3H and I*). HIV-1 sensitivity to MxB in the presence of CsA was also rescued by a second CsA escape mutant G94D (*Figure 3J*),

which is also thought to recapitulate the effect of CypA binding by stabilising the CypA binding loop. These data demonstrate that, for this particular HIV-1 isolate (HIV-1 R9), it is not CypA recruitment per se that is required for MxB sensitivity, but rather the conformational effect of CypA recruitment, likely the CypA mediated stabilisation of the CypA binding loop, which is recapitulated by the A92E and G94D mutations (*Lu et al., 2015*).

## Mapping sensitivity determinants of MxB in HIV-1 capsid

Given that nuclear entry cofactor dependence does not correlate with MxB sensitivity, we sought to understand MxB sensitivity genetically. We first tested the HIV-1 GFP constructs bearing HIV-1

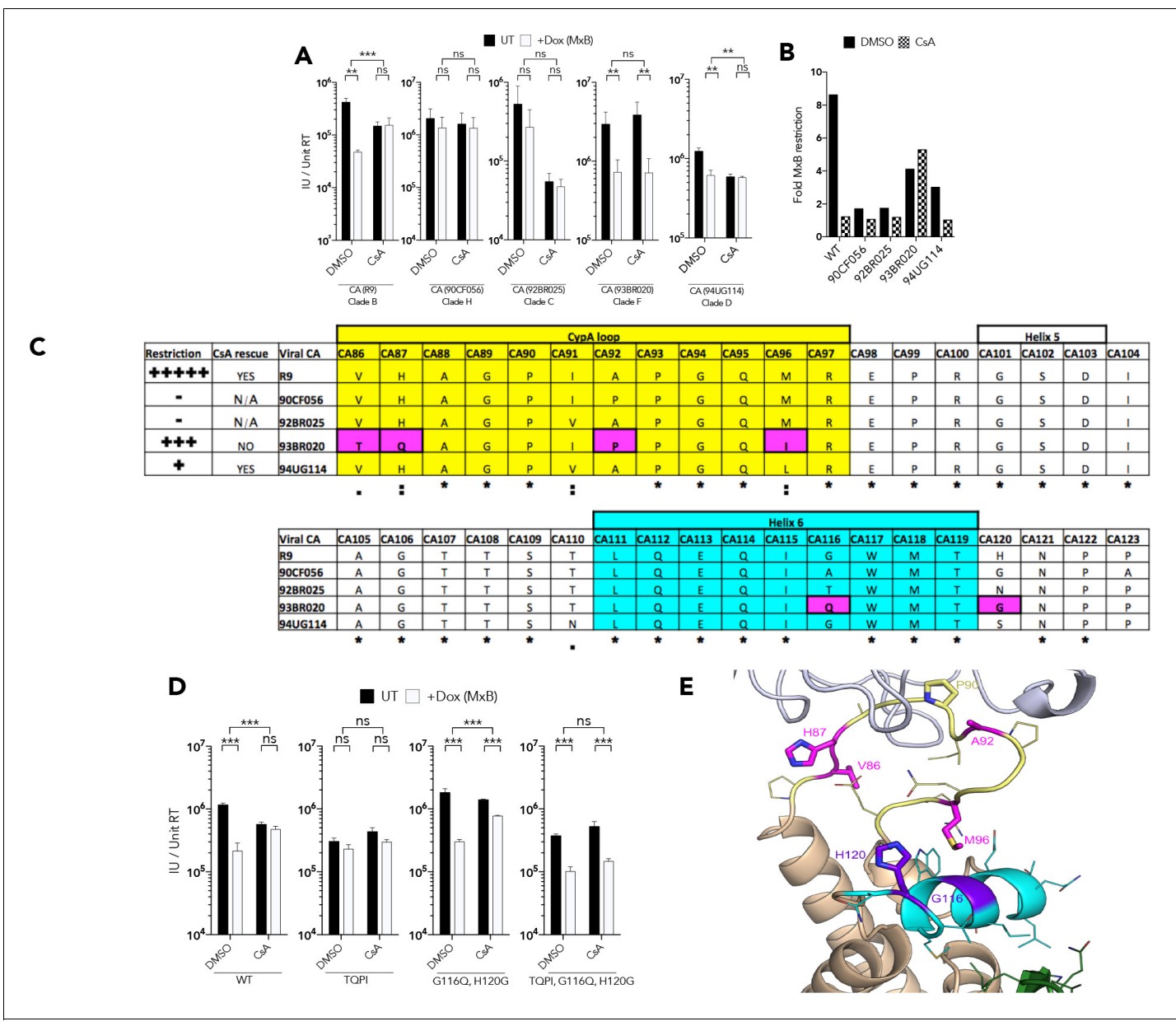

**Figure 4.** CypA induced HIV-1 capsid conformational change is required for MxB restriction. (a) Titre and (b) fold MxB restriction of HIV-1 GFP vector, containing gag-pro sequences from the stated isolates, in HEK293 cells untreated (UT) or treated with Dox to induce MxB expression (+Dox (MxB)) after CsA was added at the point of infection. (c) Summary of virus phenotypes, amino acid sequence, and conservation around the CypA binding loop of the stated isolates. (d) Titre of HIV-1-GFP wild type or indicated mutant in HEK293 cells untreated (UT) or treated with Dox to induce MxB expression (+Dox (MxB)) after CsA was added at the point of infection. (e) Structure of HIV-1 CA protein (PDB: 6ES8) indicating interaction of residues M96, G116 and H120. + / - SEM n = 3. (***) p<0.01 (**) p<0.05 (*) p<0.1 (ns) not significant p>0.1.

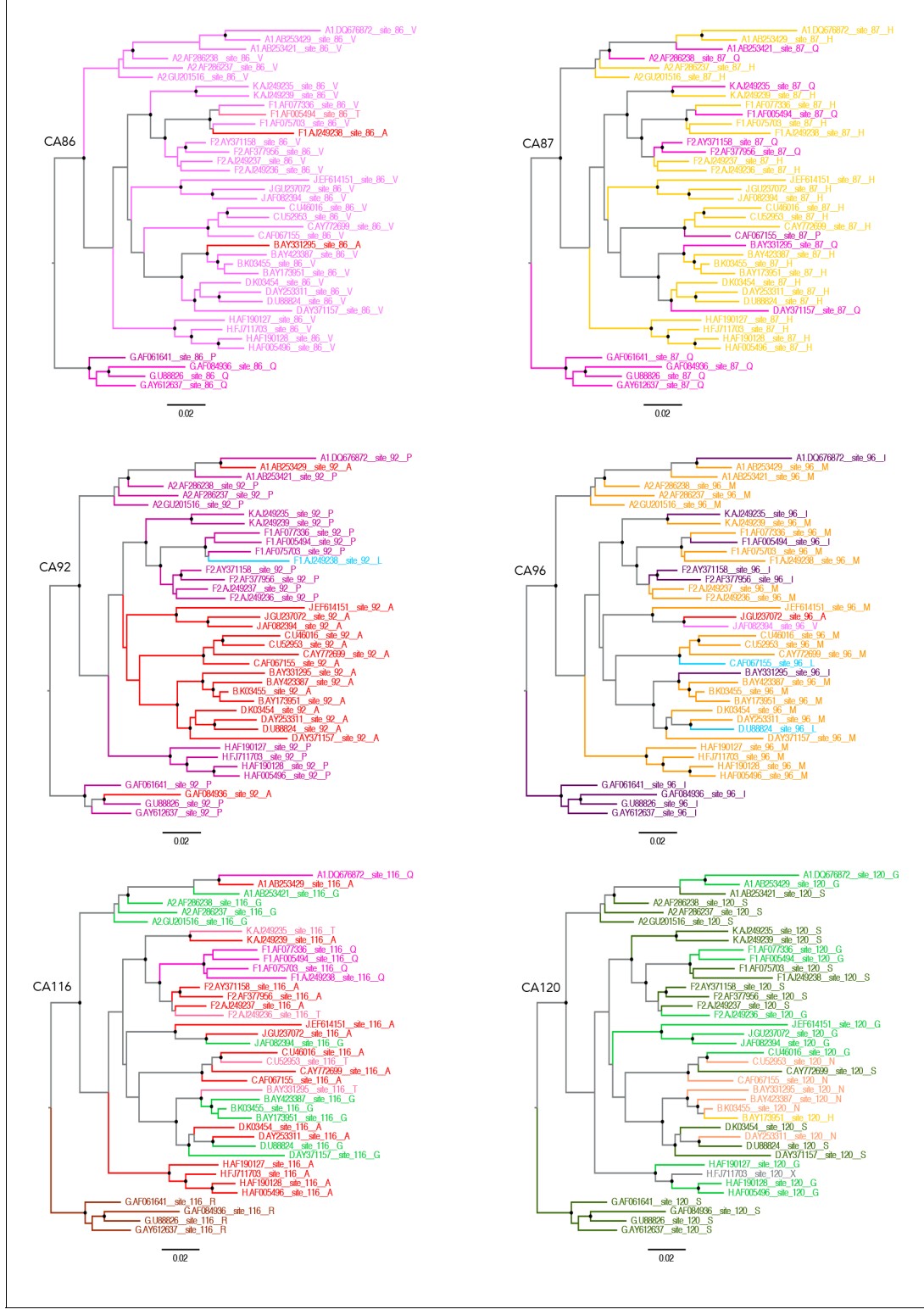

**Figure 5.** Capsid residues important for MxB restriction are variable. Phylogenetic trees of a set of HIV-1 M group subtype reference genome sequences for residues CA86, CA87, CA92, CA96, CA116 and CA120 using ChromaClade.

The online version of this article includes the following figure supplement(s) for figure 5:

**Figure supplement 1.** Phylogenetic trees of conserved and variable CA residues in HIV-1 M group viruses.

primary *gag* sequences for MxB sensitivity in the presence and absence of CsA. We found a surprising complexity of CsA/MxB sensitivities. As reported, HIV-1 GFP R9 was sensitive to MxB and CsA rescued infectivity but had itself a 2–3 fold inhibitory effect. HIV-1 GFP bearing clade H 90CF056 Gag was insensitive to MxB and CsA treatment. Clade C 92BR025 Gag was insensitive to MxB but strongly inhibited by CsA treatment. Clade F 93BR020 Gag was MxB sensitive, but in a CsA insensitive way. Clade D 94UG114 Gag was weakly sensitive to MxB and CsA rescued infection from MxB but inhibited infection by 2–3 fold (*Figure 4A,B*). The CA C-terminal region has been linked to MxB sensitivity, particularly amino acids 207–210 (*Busnadiego et al., 2014*). Indeed, a threonine at position 210 correlated with MxB sensitivity and a serine correlated with MxB escape (*Supplementary file 1*). There was however, no obvious association between the C-terminal CA sequence and sensitivity to CsA.

To gain further insight into the role of CypA in MxB sensitivity, we sought to convey the CsA insensitivity of clade F 93BR20 to the HIV-1 R9 GFP, while maintaining MxB sensitivity, by CA mutation. Inspection of the CypA binding loop sequence revealed that the critical G89 and P90 residues required for CypA recruitment are conserved but positions CA 86, 87, 92 and 96 are polymorphic between HIV-1 R9 and 93BR020 Gags (*Figure 4C*). To establish their contribution to MxB sensitivity we made the HIV-1 R9 GFP quadruple mutant V86T, H87Q, A92P M96I (named 'TQPI') to mimic the 93BR020 sequence in the CypA binding loop in CA and measured TQPI sensitivity to inhibition by CsA and MxB (*Figure 4D*). We found HIV-1 CA TQPI lost both MxB and CsA sensitivity and thus had a phenotype that was different to the parental R9 and 93BR020 Gags but similar to 90CF056 and 94UG114 Gags. Further examination of HIV-1 CA structures indicated that residue 96 is very close in space to another pair of polymorphic residues at positions 116 and 120 in the short alpha helix on the surface of the CA (*Figure 4E*). We therefore added G116Q and H120G to the HIV-1 R9 GFP CA TQPI to make HIV-1 GFP CA TQPIQG and again tested sensitivity to restriction by MxB and CsA (*Figure 4D*). We found that this virus behaved like the 93BR020 Gag bearing virus and was sensitive to MxB and this was not affected by CypA inhibition with CsA. Importantly, the double mutation HIV-1 R9 CA G116Q, H120G was not particularly different from the wild type HIV-1 R9 in either sensitivity to MxB or CsA indicating that the TQPI mutations are required to achieve the 93BR029 phenotype.

Examination of CA sequence variation in a large (*Figure 5—figure supplement 1*) or smaller (*Figure 5*) selection of HIV-1(M) sequences using Chromaclade (*Monit et al., 2019*) illustrated that five of six CA positions responsible for the different MxB and CsA sensitivity in *Figure 4* are highly variable, as opposed to the residues required for CypA recruitment at positions 89 and 90, which are completely conserved (*Figure 5* and *Figure 5—figure supplement 1*). Intriguingly, some positions, particularly 116 and 120 appear to have some degree of clade specificity. Rendering CA crystal

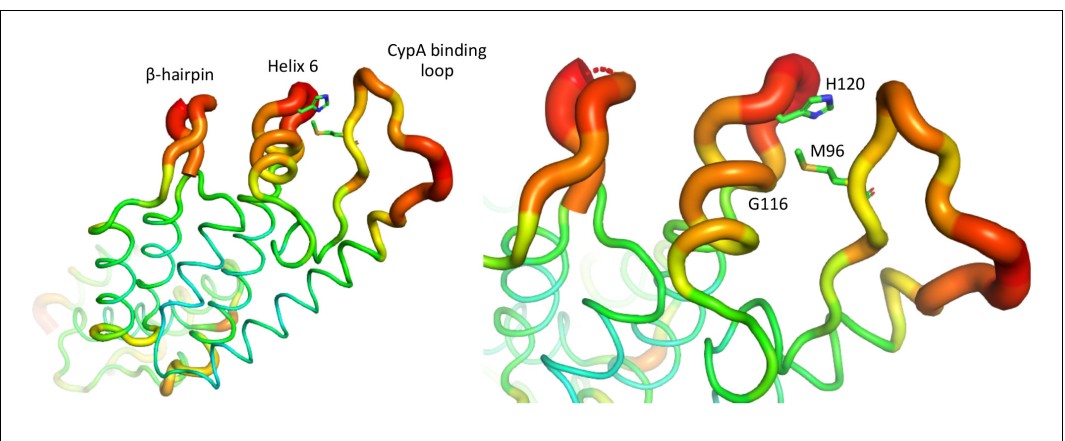

**Figure 6.** HIV-1 capsid is dynamic and variable. B factor putty representation of HIV-1 CA protein (PDB: 4XFX) indicating location of mutated residues, CypA binding loop and β-hairpin.

The online version of this article includes the following figure supplement(s) for figure 6:

**Figure supplement 1.** Location of N74 and putative MxB binding residues.

structures according to their atomic displacement parameters (B-factors) revealed that the regions around 116, 120 and 96 are particularly mobile relative to the rest of the protein (*Figure 6*). B-factors reflect the fluctuation of atoms about their average positions and provide important information about protein dynamics. Thus these structures suggest that the highly variable residues that influence MxB and CsA sensitivity are dynamic with 120, 116 and 96 close in space and likely influencing each other. NMR studies have suggested that CypA, which influences some cases of MxB sensitivity, also influences CA by changing surface dynamics (*Lu et al., 2015*). This observation, and the relatively large B-factors in CA structures for specific polymorphic residues which dictate MxB sensitivity, support the notion that MxB sensitivity is principally influenced by CA surface conformation and dynamics (*Figure 6*).

## Discussion

The reported localisation of the MxB active form to the nuclear periphery, the inhibition of HIV-1 nuclear transport by MxB and the influence on MxB sensitivity of mutations that effect nuclear transport cofactor use suggested that nuclear transport mechanism may dictate MxB sensitivity. In such a gatekeeper hypothesis we reasoned that only HIV-1 utilising a specific set of nuclear entry cofactors might be sensitive to MxB restriction because viruses using alternate nuclear entry cofactors may not encounter MxB. However, while cofactor depletion sensitivity was conserved, MxB sensitivity was not (*Figure 1*). Furthermore, HIV-1 bearing the CPSF6 binding mutant N74D was MxB insensitive, but CPSF6 depletion, which has a similar effect on HIV-1 as N74D mutation in terms of cofactor dependence and integration targeting (*Achuthan et al., 2018*; *Lee et al., 2010*; *Sowd et al., 2016*), did not render HIV-1 R9 GFP MxB insensitive. Direct recruitment of wild type CA, and not CA N74D, by MxB is the simplest explanation for these data. Concordantly, recent work (*Smaga et al., 2019*) has suggested that MxB may bind at the tri-hexamer interface to a strongly negatively charged patch composed of E71, E75 and E213 which may interact with the [11]RRR[13] motif of MxB. CA N74 lies between two of the putative MxB binding residues E71 and E75, but critically the N74 side chain extends away from the E71 and E75 side chains and into the CPSF6 binding pocket. Mutation N74D replaces the amide side chain with a negatively charged acidic side chain in close spatial proximity to the positively charged K70 (*Figure 6—figure supplement 1*). We therefore hypothesise that N74D conformationally alters this region of the capsid to prevent both CPSF6 and MxB binding.

Dependence of MxB sensitivity on CA conformation, rather than on cofactor binding, was also supported by the demonstration that mutants which mimic the conformational and dynamic changes caused by CypA recruitment to CA (*Lu et al., 2015*) (CA A92E and G94D) make HIV-1 R9 sensitive to MxB independently of CypA binding (*Figure 3*). By testing a small number of divergent Gag sequences, we illustrated that wild type HIV-1 Gag sequences have different sensitivities to both cofactors (CypA) and restriction factors (MxB). Mapping the amino acid differences required to make HIV-1 R9 behave like HIV-1 93BR020, with respect to MxB and CsA sensitivity, required changes including positions that are close together in space (*Figure 4E*), highly variable (*Figure 5*, *Figure 5—figure supplement 1*) and dynamic in X-ray crystallography (*Figure 6*) and NMR data (*Lu et al., 2015*). We assume that MxB recruitment to CA is required for restriction but it appears that the capacity to recruit CA in biochemical assays does not correlate with sensitivity to MxB restriction (*Smaga et al., 2019*). We concur with these authors, hypothesising that CA mutation can influence the inhibitory effect of MxB binding, as well as influencing whether MxB binds at all. This hypothesis is consistent with MxB insensitive HIV-1 CA mutant P90A being reported to retain MxB binding capacity. Indeed, we observe that, despite insensitivity to MxB mediated inhibition of infectivity, proviral DNA levels of HIV-1 bearing CA P90A were increased by MxB induction (*Figure 3B*). This suggests that MxB still influences HIV-1 CA P90A, possibly at the level of nuclear entry and integration, despite not reducing the number of GFP positive cells. Thus our model, based on the literature and data herein, is that MxB binds either directly or indirectly to incoming HIV-1 CA, at the nuclear rim where long MxB resides, and restriction occurs if CA is in an MxB sensitive conformation or state. Further, we conclude that CypA recruitment influences sensitivity to MxB restriction in a CA sequence sensitive way by influencing CA conformation. The effect of MxB restriction is to reduce nuclear entry of viral particles.

A recent study, demonstrating how CypA protects HIV-1 from restriction by TRIM5, likely explains inhibition of HIV-1 GFP infection by CsA in *Figure 5* (*Kim et al., 2019*). It is striking that CypA does

not protect infection from MxB but rather in some cases (R9, 94UG114) CypA causes MxB sensitivity. Thus perhaps MxB provides less selective pressure on CA than TRIM5 because MxB expression is strictly controlled by IFN, while TRIM5 expression is enhanced by IFN but typically expressed at restrictive levels in unstimulated cells (*Jimenez-Guardeño et al., 2019*). We have proposed that HIV-1 has adapted to avoid triggering IFN production (*Khan et al., 2020*; *Rasaiyaah et al., 2013*; *Sumner et al., 2019*; *Sumner et al., 2017*) and this may reduce exposure to MxB. It is striking that cofactor use is not entirely flexible. All the wild type Gag sequences tested here were dependent on Nup358 and TNPO3 for maximal infectivity and were sensitive to overexpression of mutant CPSF6 and TRIMCyp (*Figure 1*; *Ikeda et al., 2004*). The difference between absolute dependence (Nup358, CPSF6, TNPO3) versus flexible (CypA) cofactor use likely reflects their role in the infectious cycle.

Gag is under strong selective pressure from immune responses, particularly cytotoxic T lymphocytes presenting Gag derived peptides (*Carlson et al., 2015*). Thus HIV-1 must adapt to a selection of CTL escape mutations in CA and this is known to influence the relationship with specific co-factors e.g. CypA (*Brockman et al., 2007*; *Leslie et al., 2004*) and CPSF6 (*Henning et al., 2014*), and sensitivity to restriction factors e.g. MxB (*Liu et al., 2015*; *Merindol et al., 2018*; *Wei et al., 2016*) and TRIM5α (*Battivelli et al., 2011*; *Granier et al., 2013*). All HIV-1 sequences have been influenced by CTL exposure because CTL escape variants can be transmitted (*Leslie et al., 2004*). We interpret our data as suggesting a model in which HIV-1 flexibly uses CypA and changing CA sequence to influence surface dynamics and conformation to achieve maximally infectious CA behaviour. We hypothesise that CypA recruitment facilitates CA sequence change and escape from immune pressure, by providing additional conformational options to retaining insensitivity to CA targeting restriction factors while retaining appropriate interactions with essential cofactors.

Characterising restriction of HIV-1 by MxB in natural targets of HIV-1 infection such as primary T-cells and macrophages would be informative. This is challenging because inducing MxB with IFN induces the activity of other restriction factors that inhibit HIV-1 prior to MxB engagement, at the nuclear membrane. For example, the IFITM proteins inhibit HIV-1 at the level of fusion with the cell membrane (*Foster et al., 2016*) TRIM5 inhibits HIV-1 uncoating, at least in the absence of CypA activity, (*Fletcher et al., 2018*; *Kim et al., 2019*) and TRIM34 may also have anti-HIV-1 activity (*Ohainle et al., 2020*). Thus isolating MxB activity from other restriction factors is a challenge. Future experiments could use selected depletion or knock out of anti HIV-1 factors to allow comparative characterisation of MxB restriction mechanism and sensitivity determinants with results derived from cell lines, where MxB activity can be more easily isolated.

HIV-1 sensitivity to the human restriction factor MxB was originally surprising. Here we evidence a more complex picture of MxB sensitivity and cofactor dependence by using a variety of divergent wild type Gag sequences. Use of a single lab adapted strain to study host-HIV-1 interactions, here represented by the HIV-1 R9 derived packaging plasmid p8.91, which is very closely related to NL4.3 (*Zufferey et al., 1997*) is typical, and this can give the impression that HIV-1 cofactor use and restriction sensitivity is very specific. However, experimental use of more divergent HIV-1 sequences, and consideration of conservation of the amino acids under study (*Monit et al., 2019*) can give a broader perspective of functional variation in cofactor use and restriction factor sensitivity and allow development of more complex models for host-virus relationships. We expect that a deeper understanding these relationships will allow development of better antiviral therapeutics. Recent demonstration of the potency and broad antiviral activity of the HIV-1 CA-targeting Gilead Sciences new inhibitor series underlines this ambition (*Yant et al., 2019*). We also expect that a better understanding of the workings of innate immune defences, and their role in inflammatory processes, will continue to provide exciting diagnostic and therapeutic opportunities for a wide variety of diseases.

## Materials and methods

### Cell culture and reagents

Cell lines were cultured in Dulbecco's Modified Eagle's Media (DMEM) (Gibco/BRL; Paisley, UK) supplemented with 10% (v/v) fetal calf serum (FCS) and penicillin (Pen; 100 U/ml), streptomycin (Strep; 100 µg/ml) unless otherwise stated. G418 was used at 1000 µg/ml. Puromycin was used at 5 µg/ml. 293 cell lines were incubated at 37°C, 5% (v/v) $CO_2$. All other cell lines were incubated at 37°C, 10%

(v/v) $CO_2$. Inducible MxB cell lines were made by transduction of HEK293, THP-1 and A3.01 with pLVX-Tet3G (Clontech), containing the Tet-responsive transactivator, and cells were selected for one week in G418. Cells were subsequently transduced with pLVX-mCherryKilled-MxB (modified from Clontech) and selected for one week in puromycin (2 µg/ml). Single cell clones were produced by limiting dilution in 96 well plates and assayed for optimal MxB induction. MxB inducible cell lines were induced for MxB expression by addition of 2 µg/ml of doxycycline for 48 hr prior to infection. Cell lines used in this study HEK293, U87-MG, HeLa, THP-1, A3.01. Cells routinely tested negative for mycoplasma using the Lonza MycoAlert mycoplasma detection kit. Antibodies used in this study; TNPO3 (ab54353, Abcam), CPSF6 (ab175237, Abcam), MxB (sc47197 N-17, Santa Cruz Biotechnology), CypA (BML-SA296-0100, ENZO), Nup358 (C288), Tubulin (DM1A, EMD Millipore), β-actin (ab6276, Abcam).

## Plasmids and cloning

Human MxB was PCR cloned from cDNA derived from IFN-induced THP-1 cells and cloned into pLVX-mCherryKilled vector at a multiple cloning site under the control of a doxycycline-inducible promoter. MxB mutant derivatives were produced by QuickChange site-directed mutagenesis and sub-cloned between appropriate restriction enzyme sites. All cloning and mutagenesis was confirmed by sequencing.

## Viral infectivity assay

Cells were seeded at $1 \times 10^5$/ml and 1 ml/well in 24 well plates; after 24 hr media was replaced with 250 µl of media containing the appropriate concentration of virus. After 48 h cells were thoroughly trypsinised with Trypsin + 0.25% EDTA (100 µl) and fixed by addition of 1X PBS + 4% paraformaldehyde (PFA; 100 µl). All experiments included an uninfected control. Each virus was titrated on each cell line initially starting with neat supernatant followed by a 10–12 point 3-fold dilution; subsequent experiments contained at least four viral dilutions points with the aim of achieving a 1–30% infection range from which titres were calculated. Titres were normalised to units of reverse transcriptase (UnitRT) in viral supernatant as calculated by SG-PERT (*Pizzato et al., 2009*) or RT ELISA.

## Lentiviral production

HEK293T cells were routinely passaged 1:4 in DMEM supplemented with 10% (v/v) FCS and Pen/Strep. HEK293T cells were seeded (DMEM + 10% FCS + 1% Pen/Strep) at a cell density of $4 \times 10^6$ in 10 cm cell culture dishes. Fugene-6 transfectant reagent (Promega; 10 µl) was incubated at room temperature with Optimal Minimum Essential Media (200 µl; Opti-MEM) for 5mins. Transfection DNA mix was prepared containing 1 µg of viral *gag-pol* (p8.91), 1 µg of VSV-G *env* (pMDG) and 1.5 µg of GFP-encoding genome (pCSGW) or expression plasmid to a total volume of 15 µl, added to the Fugene-6 mix and incubated for 30mins (*Fletcher et al., 2015*). Cell media was replaced and the transfectant mix was added drop-wise and incubated for 24 hr. Culture media was replaced (8 ml; DMEM 10% FCS) and virus-containing media was harvested 48 hr and 72 hr post-transfection, passed through 0.44 µm Nalgene sterilization filter, aliquoted and stored at −80˚C.

## Short hairpin RNA knockdown

Cells were plated at $4 \times 10^5$ cells/well in 2 ml in 6-well plates prior to transduction with appropriate short hairpin RNA, which have been described previously (*Rasaiyaah et al., 2013*; *Schaller et al., 2011*), cloned into the retroviral expression vector pC-SIREN RetroQ (Clontech). Cell media was replaced 24 hr post-transduction and cells stimulated with dox (2 µg/ml). After 24 hr, cells were seeded at $1 \times 10^5$/ml/well in 24 well plates for viral infectivity assays.

## Immunofluorescence confocal microscopy

Cells were plated in 24 well plates at a density of $5 \times 10^4$ on glass cover slips, previously washed in EtOH and then PBS. After ~24 hr, cells were fixed with paraformaldehyde (4%) for 15 min and washed with PBS (1 ml). Cover slips were transferred to a fresh 24-well plate and permeabilised with PBS/FCS (2%)/Triton X-100 (0.5%) for 30 min after which the cells were washed with PBS (1 ml) for 5 min. Permeabilised cells were incubated at room temperature for 1 hr with the appropriate dilution of primary antibody in PBS/FCS (2%) then washed three x 5 min with PBS. Cells were incubated with

the appropriate anti-species fluorophore-conjugated secondary antibody for 1 hr in the dark after which cells were washed three x 5 min with PBS. Cover slips were mounted on glass slides using VectorShield containing DAPI nuclear stain and stored in the dark at 4°C. Cells were visualized using the Lucia confocal laser microscope.

## Immunoblots

Resolving acrylamide gels were prepared as follows: acrylamide/bis-acrylamide (7.5–12%), Tris (1M, pH 8.8; 25.2%), sodium dodecylsulphate (SDS; 0.1%), ammonium persulfate (APS; 0.05%), tetramethylethylenediamine (TEMED; 0.05%). Stacking acrylamide gels were prepared as follows: acrylamide/bis-acrylamide (4%), Tris (0.5M, pH 6.8; 25.2%), sodium dodecylsulphate (SDS; 0.1%), ammonium persulfate (APS; 0.05%), tetramethylethylenediamine (TEMED; 0.1%). Cells were collected by centrifugation and resuspended in SDS loading buffer supplemented with 2-mercaptoethanol (5%) and heated to 95°C for 10 min to fully denture the proteins. Cell extract samples and protein markers were run through acrylamide gels at 120V. Proteins were transferred by electrophoresis on to nitrocellulose membranes that were subsequently blocked with phosphate-buffered saline plus Tween (1%; PBS-T) supplemented with skimmed milk powder (10% w/v). Membranes were incubated with appropriate dilution of primary antibody followed by anti-species secondary antibody conjugated to horseradish-peroxidase (HRP), each diluted in PBS-T plus skimmed milk powder (10% w/v), for 1 hr at room temperature with agitation followed by washing in PBS-T ($3 \times 5$ mins) after each antibody incubation. Membranes were incubated with enhanced-chemiluminescence (ECL) solution for 5 min, exposed to photographic film and developed using an X-ray film processor.

## Quantitative PCR

To measure viral RT products, cells were collected at 6 hr post infection; for 2-LTR circles cells were collected at 72 hr post-infection; for proviral DNA cells were collected after passage for 3 weeks. DNA was isolated using QIAamp DNA Mini Blood kit (Qiagen). Viral DNA products were measured by TaqMan quantitative PCR using ABI FastCycler 9600. Viral RT products and proviral DNA was detected with the following GFP specific primer pair– fwd:5′-CAACAGCCACAACGTCTATATCAT-3′ and rev:5′-ATGTTGTGGCGGATCTTGAAG-3′ and probe – 5′-FAM-CCGACAAGCAGAA-GAACGGCATCAA-TAMRA-3′. Viral 2-LTR circles were detected with the following primer pair – fwd:5′-AACTAGAGATCCCTCAGACCCTTTT-3′ and rev:5′-CTTGTCTTCGTTGGGAGTGAATT-3′ and probe – 5′-FAM-TTCCAGTACTGCTAGAGATTTTCCACACT-TAMRA-3′. DNA copy number was determined by comparison with a standard curve made using CSGW plasmid DNA of known concentrations.

## Acknowledgements

We thank Eran Bacharach and Richard Milne for critical reading of the manuscript. This work was funded by an MRC PhD studentship to RM, a Wellcome Trust Senior Biomedical Research Fellowship to GJT and the European Research Council under the European Union's Seventh Framework Programme (FP7/2007-2013)/ERC (grant HIVInnate 339223) and the National Institute for Health Research University College London Hospitals Biomedical Research Centre. DAJ is supported by an Australian Research Council Discovery Project grant (DP180101384) and a UNSW Scientia Fellowship. GJT and DAJ are also supported by a Wellcome Trust Collaborative Award.

## Additional information

### Funding

| Funder | Grant reference number | Author |
| --- | --- | --- |
| Medical Research Council | | Richard J Miles |
| Wellcome | | Gregory J Towers |
| H2020 European Research Council | HIVInnate 339223 | Gregory J Towers |
| Australian Research Council | DP180101384 | David A Jacques |

| | |
|---|---|
| University of New South Wales | David A Jacques |
| National Institute for Health Research | Gregory J Towers |

The funders had no role in study design, data collection and interpretation, or the decision to submit the work for publication.

### Author contributions

Richard J Miles, Conceptualization, Data curation, Formal analysis, Validation, Investigation, Visualization, Methodology, Writing - original draft, Project administration, Writing - review and editing; Claire Kerridge, Laura Hilditch, Christopher Monit, Data curation, Formal analysis, Investigation; David A Jacques, Supervision, Visualization, Writing - review and editing; Greg J Towers, Conceptualization, Resources, Data curation, Formal analysis, Supervision, Funding acquisition, Validation, Visualization, Methodology, Writing - original draft, Project administration, Writing - review and editing

### Author ORCIDs

David A Jacques ![ORCID] https://orcid.org/0000-0002-6426-4510
Greg J Towers ![ORCID] https://orcid.org/0000-0002-7707-0264

### Decision letter and Author response

Decision letter https://doi.org/10.7554/eLife.56910.sa1
Author response https://doi.org/10.7554/eLife.56910.sa2

## Additional files

### Supplementary files

- Supplementary file 1. Sequence alignment of HIV-1 isolates between aa207-210 of capsid protein.
- Transparent reporting form

### Data availability

All data generated or analysed during this study are included in the manuscript and supporting files.

The following previously published dataset was used:

| Author(s) | Year | Dataset title | Dataset URL | Database and Identifier |
|---|---|---|---|---|
| Gres AT, Kirby KA, KewalRamani VN, Tanner JJ, Pornillos O, Sarafianos SG | 2015 | Structure of the native full-length HIV-1 capsid protein | https://www.rcsb.org/structure/4xfx | RCSB Protein Data Bank, 4XFX |

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
