## [Decision Letter]

**Acceptance summary:**

This paper demonstrates that the sensitivity of HIV-1 to the host antiviral factor MxB is dictated by the conformation of the viral capsid, rather than the viral nuclear entry cofactor use. The interaction of the viral capsid with cylcophilin A plays an important role in the sensitivity of the virus to MxB. HIV-1 primary isolates have different MxB sensitivities due to differences in Gag sequence but similar cofactor dependencies, suggesting a complex relationship between cyclophilin dependence and MxB sensitivity likely driven by CTL escape.

**Decision letter after peer review:**

Thank you for submitting your article "MxB sensitivity of HIV-1 is determined by a highly variable and dynamic capsid surface" for consideration by *eLife*. Your article has been reviewed by two peer reviewers, and the evaluation has been overseen by a Reviewing Editor and Päivi Ojala as the Senior Editor. The reviewers have opted to remain anonymous.

The reviewers have discussed the reviews with one another and the Reviewing Editor has drafted this decision to help you prepare a revised submission.

Summary:

In this paper, the authors analyzed the phenotypes of HIV-1 CA mutants for their sensitivity to MxB and dependence on nuclear entry confactors. They demonstrated that the sensitivity of HIV-1 to MxB is not determined by the viral nuclear entry cofactor use. They further provided evidence indicating that the interaction of CA with cyclophilin A is important for MxB sensitivity, suggesting that the conformation of CA surface dictates the sensitivity. These are interesting findings that should help to better understand the mechanism of action of MxB. The paper fits the scope of *eLife*, though some concerns need to be addressed before publication.

Essential revisions:

1) This paper clearly showed that the sensitivity of HIV-1 CA to MxB is not determined by the nuclear entry cofactor use. However, it is not clear exactly what determines the sensitivity. It seems that the authors tried to avoid stating that the sensitivity is dictated by the interaction of CA with MxB, as some MxB escape CA mutants were reported to interact with MxB in in vitro assays. The term "capsid conformation and dynamics" used in this paper is vague in describing the mechanism of MxB action. The two MxB escape CA mutants, P90A and N74D were analyzed in details. While they are both insensitive to MxB, P90A was shown to bind MxB (Smaga et al., 2019) and N74D was proposed by the authors to prevent MxB binding based on the model described in the same Smaga paper (Discussion paragraph one). If the authors wish to conclude that MxB sensitivity is dependent on HIV-1 capsid binding to MxB, the authors need to test experimentally whether the N74D mutant binds MxB. Otherwise, the authors need to modify their claims. In either case, the authors need to describe their working model more clearly.

2) HIV-1 nuclear entry is dictated and influenced by both viral and cellular factors. Logically, dependence on a specific cellular factor(s) does not equate the same nuclear entry mechanism. For example, two different signaling cascades or nuclear entry pathways may happen to use the same cellular factor, but MxB may hit one pathway but not the other. The data support the conclusion that MxB sensitivity of a specific HIV-1 Gag sequence is independent of a particular factor, but may not be necessarily correct when extending the conclusion to the nuclear import pathway dictated by this Gag. The authors were careful with their wording in the Abstract and Discussion. But in the Results section, sometimes "nuclear entry routes" or "nuclear entry mechanisms" are used. The authors need to describe their results more carefully.

---

## [Author Response]

Essential revisions:1) This paper clearly showed that the sensitivity of HIV-1 CA to MxB is not determined by the nuclear entry cofactor use. However, it is not clear exactly what determines the sensitivity. It seems that the authors tried to avoid stating that the sensitivity is dictated by the interaction of CA with MxB, as some MxB escape CA mutants were reported to interact with MxB in in vitro assays. The term "capsid conformation and dynamics" used in this paper is vague in describing the mechanism of MxB action. The two MxB escape CA mutants, P90A and N74D were analyzed in details. While they are both insensitive to MxB, P90A was shown to bind MxB (Smaga et al., 2019) and N74D was proposed by the authors to prevent MxB binding based on the model described in the same Smaga paper (Discussion paragraph one). If the authors wish to conclude that MxB sensitivity is dependent on HIV-1 capsid binding to MxB, the authors need to test experimentally whether the N74D mutant binds MxB. Otherwise, the authors need to modify their claims. In either case, the authors need to describe their working model more clearly.

We agree that our manuscript is improved by a clearer discussion of these points. We have added the following text to the Discussion:

“We assume that MxB recruitment to CA is required for restriction but it appears that the capacity to recruit CA in biochemical assays does not correlate with sensitivity to MxB restriction (Smaga et al., 2019). We concur with these authors, hypothesising that CA mutation can influence the inhibitory effect of MxB binding, as well as influencing whether MxB binds at all. This hypothesis is consistent with MxB insensitive HIV-1 CA mutant P90A being reported to retain MxB binding capacity. Indeed, we observe that, despite insensitivity to MxB mediated inhibition of infectivity, proviral DNA levels of HIV-1 bearing CA P90A were increased by MxB induction (Figure 3B). This suggests that MxB still influences HIV-1 CA P90A, possibly at the level of nuclear entry and integration, despite not reducing the number of GFP positive cells. Thus our model, based on the literature and data herein, is that MxB binds either directly or indirectly to incoming HIV-1 CA, at the nuclear rim where long MxB resides, and restriction occurs if CA is in an MxB sensitive conformation or state. Further, we conclude that CypA recruitment influences sensitivity to MxB restriction in a CA sequence sensitive way by influencing CA conformation. The effect of MxB restriction is to reduce nuclear entry of viral particles.”

2) HIV-1 nuclear entry is dictated and influenced by both viral and cellular factors. Logically, dependence on a specific cellular factor(s) does not equate the same nuclear entry mechanism. For example, two different signaling cascades or nuclear entry pathways may happen to use the same cellular factor, but MxB may hit one pathway but not the other. The data support the conclusion that MxB sensitivity of a specific HIV-1 Gag sequence is independent of a particular factor, but may not be necessarily correct when extending the conclusion to the nuclear import pathway dictated by this Gag. The authors were careful with their wording in the Abstract and Discussion. But in the Results section, sometimes "nuclear entry routes" or "nuclear entry mechanisms" are used. The authors need to describe their results more carefully.

This is an excellent point. We must admit that, throughout this process, we have thought of cofactor dependency as dictating HIV-1 nuclear entry mechanism. But the reviewers are absolutely right; we do not know whether the actual mechanism of nuclear entry is changing as we manipulate Gag sequence or cofactor availability. We agree, that our manuscript is therefore clearer if we avoid the phrases “nuclear entry routes” or “nuclear entry mechanisms” when discussing the results. This is particularly true, because in the end, we conclude that nuclear entry route or mechanism do not impact MxB sensitivity. So it is indeed clearer not to bring this up this complex point in the first place.

We have gone through the Results section and altered “nuclear entry routes” or “nuclear entry mechanisms” to more specifically reflect the data in the text as follows:

“Given that only the nuclear periphery-localising long form of MxB restricts HIV-1 infection, and that MxB apparently restricts HIV-1 nuclear entry, we hypothesised that MxB sensitivity of the different Gag sequences might be influenced by their nuclear entry route or mechanism. This notion is consistent with published demonstration that HIV-1 CA mutants thought to use different nuclear entry mechanisms are insensitive to MxB restriction.” changed to “Given that only the nuclear periphery-localising long form of MxB restricts HIV-1 infection, and that MxB apparently restricts HIV-1 nuclear entry, we hypothesised that MxB sensitivity of the different Gag sequences might be influenced by their dependence on cofactors associated with nuclear entry. This notion is consistent with published demonstration that HIV-1 CA mutants, which have altered sensitivity to nuclear entry associated cofactor depletion, are insensitive to MxB restriction.”

“To probe the relationship between nuclear entry mechanism and MxB sensitivity”

changed to “To probe the relationship between cofactor use and MxB sensitivity”

“To further probe the role of nuclear entry mechanism in determining MxB restriction” changed to “To further probe the mechanism of MxB restriction”

“Together, these observations suggest that HIV-1 bearing CA N74D is insensitive to MxB as a direct result of the CA mutation and not due to altered nuclear entry mechanism, i.e. cofactor or pathway use.” changed to “Together, these observations suggest that HIV-1 bearing CA N74D is insensitive to MxB as a direct result of the CA mutation and not due to loss of CPSF6 recruitment.”

“But HIV-1 P90A also has reduced sensitivity to depletion of Nup358, TNPO3 and CPSF6 and has retargeted proviral integration preferences, which again suggests a connection between nuclear transport mechanism and MxB sensitivity (Achuthan et al., 2018; Lee et al., 2010; Schaller et al., 2011; Sowd et al., 2016).” changed to “But HIV-1 P90A also has reduced sensitivity to depletion of Nup358, TNPO3 and CPSF6 and has retargeted proviral integration preferences, which again suggests a connection between nuclear transport and MxB sensitivity (Achuthan et al., 2018; Lee et al., 2010; Schaller et al., 2011; Sowd et al., 2016).”

“Together, these data are consistent with CypA either acting as a cofactor for MxB interaction with the HIV-1 capsid or MxB targeting a specific nuclear import mechanism accessed by HIV-1 through CypA recruitment to the viral capsid.” changed to “Together, these data are consistent with CypA either acting as a cofactor for MxB interaction with the HIV-1 capsid or MxB targeting a form or conformation of Gag that is CypA dependent.”

“Given that nuclear entry mechanism does not correlate with MxB sensitivity” changed to “Given that nuclear entry cofactor dependence does not correlate with MxB sensitivity”